# *Arabidopsis Toxicos en Levadura 12* (*ATL12*): A Gene Involved in Chitin-Induced, Hormone-Related and NADPH Oxidase-Mediated Defense Responses

**DOI:** 10.3390/jof7100883

**Published:** 2021-10-19

**Authors:** Feng Kong, Tingwei Guo, Katrina M. Ramonell

**Affiliations:** 1Department of Biological Sciences, The University of Alabama, Tuscaloosa, AL 35401, USA; fkong@crimson.ua.edu; 2Center for Craniofacial Molecular Biology, University of Southern California, Los Angeles, CA 90089, USA; tingweig@usc.edu

**Keywords:** chitin, fungal defense, plant hormone, NADPH oxidase, *Arabidopsis thaliana*, *Golovinomyces cichoracearum*

## Abstract

Plants, as sessile organisms, have evolved complex systems to respond to changes in environmental conditions. Chitin is a Pathogen-Associated-Molecular Pattern (PAMP) that exists in the fungal cell walls, and can be recognized by plants and induce plant pattern-triggered immunity (PTI). Our previous studies showed that *Arabidopsis Toxicos en Levadura 12 (ATL12)* is highly induced in response to fungal infection and chitin treatment. We used the model organism *Arabidopsis thaliana* to characterize *ATL12* and explore its role in fungal defense. Histochemical staining showed that *pATL12*-GUS was continually expressed in roots, leaves, stems, and flowers. Subcellular co-localization of the ATL12-GFP fusion protein with the plasma membrane-mcherry marker showed that ATL12 localizes to the plasma membrane. Mutants of *atl12* are more susceptible to *Golovinomyces* *cichoracearum* infection, while overexpression of *ATL12* increased plant resistance to the fungus. *ATL12* is highly induced by chitin after two hours of treatment and *ATL12* may act downstream of *MAPK* cascades. Additionally, 3,3′-diaminobenzidine (DAB) staining indicated that *atl12* mutants generate less reactive oxygen species compared to wild-type Col-0 plants and RT-PCR indicated that *ATL12-*regulated ROS production may be linked to the expression of *respiratory burst oxidase homolog protein D/F* (*AtRBOHD/F*). Furthermore, we present evidence that *ATL12* expression is upregulated after treatment with both salicylic acid and jasmonic acid. Taken together, these results suggest a role for *ATL12* in crosstalk between hormonal, chitin-induced, and NADPH oxidase-mediated defense responses in Arabidopsis.

## 1. Introduction

In nature, plants are exposed to various environmental stresses such as pathogen attack (fungi, bacteria, insects, etc.) and abiotic stress (heat, salt, drought, etc.). Such environmental stressors are common conditions that directly affect plant metabolism, resulting in changes in gene expression and altering plant growth and development [1]. Due to climate change and global food distribution patterns, fungal pathogens pose an increased threat to current and future food security [2]. Worldwide, there are over 500 powdery mildew species that colonize over 10,000 different plant species and are the cause of significant crop losses, especially to plants in the *Cucurbitaceae* (cucurbits) and *Vitaceae* (grape) families [3,4]. Once disease develops, large numbers of asexual spores are formed, spreading the infection to the whole plant and then via wind dispersion to surrounding plants [4]. As biotrophic pathogens, powdery mildews acquire nutrients from their host plants through a specialized feeding structure, the haustoria. The redirection of photosynthate and other critical resources to the spreading fungal infection results in a reduction in overall crop yield and losses in fruit and/or vegetable product quality [3,5]. By elucidating the signaling transduction mechanisms initiated upon powdery mildew infection, practical solutions may be uncovered for the control of plant fungal disease in crops. [5,6].

Although plants face numerous pathogen attacks in nature, they have evolved efficient immune responses to protect themselves from disease. Generally, pathogens are recognized by Pathogen-Associated-Molecular Patterns (PAMPs) in plants [5]. The detection of PAMPs by plant pattern recognition receptors (PRR), induces a defense response called pattern-triggered-immunity (PTI) [1]. There are several different types of PAMPs, including bacterial flagellin [7], chitin and its derivatives [8], the secretory protein OPEL [9], and bacterial elongation factor 2 [5]. These PAMPs are recognized by their corresponding PRRs, including the Arabidopsis flagellin sensitive 2 (FLS2) receptor, LysM-domain-containing receptor-like kinase 1 (RLK1)/chitin–elicitor receptor kinase 1 (CEBiP), and the Arabidopsis EF-Tu receptor [5,6]. After a pathogen is recognized, a number of defense mechanisms are induced, such as the closure of leaf stomata, the generation of reactive oxygen species [10], the hypersensitive response (HR), and the production of defense-related proteins [7].

Chitin, a polysaccharide that is composed of ß-1-4-linked N-acetylglucosamine, is a PAMP, which exists in fungal cell walls and insect exoskeletons [11]. Plants do not contain chitin, but they do produce chitin degrading enzymes (chitinases) that can digest fungal cell walls leading to the release of small chitin fragments that then function as PAMPs, eliciting a defense response [9]. In Arabidopsis, LysM-receptor-like kinases *AtCERK1*, and *AtLYK5/AtLYK6* form a chitin receptor complex, which are able to detect chitin and induce chitin-triggered immunity [12,13,14,15]. In rice, during fungal infection, chitin fragments are detected by CEBiP (chitin elicitor binding protein), a LysM domain-containing protein [12,13,14,15]. CEBiP does not have an intracellular domain, so the recognition of chitin clearly required the assistance of a “partner” receptor-like kinase or receptor-like protein [14]. In subsequent work, the LysM RLK1/chitin-elicitor receptor kinase 1 (OsCERK1) complex was found to play a critical role in chitin recognition [16]. After chitin recognition, both *AtCERK1*/*AtLYK5/AtLYK6 complex and* the OsCERK1/CEBiP complex are able to activate a MAPK cascade and induces the expression of several transcription factors downstream in the chitin signaling pathway. These transcription factors regulate the expression of defense-related genes, leading to disease resistance [4,13]. Based on the latest evidence, it seems that the chitin signaling pathway and the bacterial flagellin and EF-Tu induced signaling pathways all share common downstream elements [14]. The recognition of all three elicitors: chitin, bacterial flagellin, and elongation factor-Tu induce the same MAPK kinase cascade and leads to the expression of WRKY transcriptional factors and other defense-related proteins [17].

The Ubiquitin/26S proteasome system plays an important role in degrading misfolded proteins and in regulating signals in plant innate immunity [17]. Additionally, the Ubiquitin/26S proteasome system is involved in many significant processes, including cell division, plant growth and development, plant-pathogen interactions, plant hormone responses, and biotic and abiotic stress resistance (cold, salt, and drought) [18,19]. In the Arabidopsis genome 2 E1 ubiquitin-activating enzymes, 37 E2 ubiquitin-conjugating enzymes, and 1415 E3 ubiquitin ligases are encoded [11]. Generally, E3 ubiquitin ligases can be divided into two main groups, based on their specific structure. The first are the Homology to E6-AP C-terminus (HECT) domain-containing E3 Ubiquitin ligases, containing seven distinct gene categories. The second group is the really-interesting-new gene/U-box domain-containing (RING) E3 ubiquitin ligases, which can either function as ubiquitin ligases alone or in the form a complex [20]. Both types of E3 ligases play critical roles in plant innate immunity [21]. For example, the *Arabidopsis Toxicos en Levadura 9* (*ATL9*) gene, which encodes a conserved RING-H2 finger protein that has E3 ubiquitin ligase activity, has been shown to play an important role in plant elicitor-mediated signaling [11]. Also, *Arabidopsis Toxicos en Levadura 78* (*ATL78*) has been shown to mediate signaling in drought stress responses [22]. The *Arabidopsis Toxicos en Levadura 2* (*ATL2*) gene family has 13 members, and all appear to encode conserved RING-Zinc finger proteins with potential E3 ubiquitin ligase activity. They have been shown to be activated by chitin and play important roles in plant defense pathways [22]. The RING-finger domain in *ATL2* family members consists of a Cys3HisCys4 amino acid motif, which binds two zinc cations. This protein domain can range from 40 to 60 amino acids in length and is known to be essential for E3 ligase activity [11,22]. All 13 members of the *ATL2* family contain both RING domains and transmembrane domains. Four ATL2 members contain PEST domains, which are known to be associated with proteins that have short half-lives within the cell [22]. Our preliminary study has determined that loss-of-function mutations in the *atl12* gene led to increased susceptibility to the powdery mildew pathogen *Golovinomyces cichoracearum*.

In this work, we use the model plant *Arabidopsis thaliana* to determine the function of *ATL12* in chitin elicitor-triggered defense signaling and discover ATL12’s mode of activation and regulation. We found that loss of function mutants in *atl12* are more susceptible to the powdery mildew infection while overexpression of *ATL12* in Arabidopsis increased resistance to mildew infection. Histochemical staining showed that a promoter construct of *ATL12* linked to beta-glucoronidase (*pATL12-GUS*) was continually expressed in Arabidopsis roots, leaves, stems, and flowers. Subcellular co-localization of the ATL12-GFP fusion protein with the plasma membrane-mcherry marker showed that ATL12 localizes to the plasma membrane. *ATL12* is highly induced by chitin at early stages (two hours) after chitin treatment and this expression is linked to MAPK cascade activation. Additionally, the expression of respiratory burst oxidase homolog protein D/F (*AtRBOHD/F*) is decreased in the *atl12* mutant, while the expression of *ATL12* is not affected in *atrbohd* and *atrbohf* mutants. This suggests that chitin-induced *ATL12* expression is also linked to NADPH oxidase *AtRBOHD/F*-driven ROS production. Furthermore, the expression of *ATL12* is upregulated after treatment with salicylic acid (SA) and jasmonic acid (JA), which suggest a possible role for *ATL12* in plant hormone-mediated defense responses. Taken together, these results indicate that *ATL12* is involved in crosstalk between the SA-, JA-, chitin-induced MAPK and NADPH oxidase-mediated defense responses in Arabidopsis.

## 2. Materials and Methods

### 2.1. Biological Materials

*Arabidopsis thaliana* ecotype Columbia (Col-0) was used as control in all experiments. T-DNA insertional mutants of *atl12:* (SALK_201056C (*atl12*), SALK_066923C (atl12a), SALK_0950303C (atl12b)), *cerk1* (SALK_007193C, AT3G21630), *mapk3* (CS349836, AT3G45640), *wrky53* (SALK_034157C, AT4g23810), *atrbohd* (SALK_083046), and *atrbohf* (SALK_034668) were obtained from the Arabidopsis Biological Resource Center (ABRC, Ohio State University, Columbus, OH, USA). To screen for homozygous T-DNA mutants, PCR reactions were performed using the following primers:

*atl12* (LP 5′-CAATCACCAATCACCTCCAAG-3′, RP 5′AATCACAGGATCGTTGTTTCG-3), *atl12a* (LP 5′-GTATTGACCAGTGGCTTGAGC-3′, RP 5′-GCAGCTTTAGTGGCGTACATG-3′), *atl12b* (LP 5′-TAATCTCGCGAATTCATCACC-3′, RP 5′- GTCAAGCGACAGATTTTCGTC-3′), *cerk1* (LP 5′-CAATTGGTCACTGCAACATTG-3′, RP 5′- TTGTACCTGAGGATTGGATCG-3′), *wrky53* (LP 5′-TCAGGCACGACTTAGAGAAGC-3′, RP 5′-GGGAAAGTTGTGTCAATCTCG-3′), *mapk3* (LP5′-TCTGCCTTTCCCTCTTCTCTC-3′, RP 5′-GACAGCATTGACTCTGGCTCT-3′), *atrbohd* (LP 5′ATCAGTGCCGCATATTCTTTG-3′, RP 5′ ATCTTTCTTCCGAAGCACC TC-3′), *atrbohf* (LP 5′-AAACCAACACGCACCTTATTG-3′, RP 5′-ATGAAATTGGCATTGCATTTC-3′) and T-DNA insertion border primer LBb1.3: 5′- ATTTTGCCGATTTCGGAAC-3′).

The genomic DNA of four-week-old seedlings was extracted using the Cetyltrimethylammonium bromide (CTAB) method. Two paired PCR reactions (FP+RP and LB+RP) and gel electrophoresis were used to screen for *ATL2* homozygous lines. All plants were grown in a growth chamber under controlled conditions at 22 °C day/19 °C night with 16 h of light per 24 h and 50% humidity. *Golovinomyces cichoracearum* strain UCSC1 was used as the fungal pathogen and propagated in cucumber. *G.cichoracearum* was maintained at 22 °C day/19 °C night with 16 h of light per 24 h and 85% relative humidity.

### 2.2. Bioinformatics Analysis

The complete sequences of *ATL2* were obtained from GenBank, NCBI, and sequences were compared using the Blastx and Blastn algorithms (http://blast.ncbi.nlm.nih.gov/Blast.cgi) (ATL12 accessed on 14 April 2019). Open-reading frames (ORFs) of ATL12 were analyzed with ORF Finder (http://www.ncbi.nlm.nih.gov/gorf/orFig.cgi) (accessed on 14 April 2019), and its gene structure and functional domains were predicted with Uniprot (https://www.uniprot.org) (accessed on 14 April 2019) and Smart (http://smart.embl-heidelberg.de/) (accessed on 14 April 2019). The amino acid sequences and functional domains of ATL2 were identified with Blastp (http://blast.ncbi.nlm.nih.gov/Blast.cgi) (accessed on 14 April 2019). Sequence alignments were made using T-coffee (http://tcoffee.crg.cat/apps/tcoffee/do:regular) (accessed on 14 April 2019) and Boxshade (http://www.ch.embnet.org/software/BOX_form.html) (accessed on 14 April 2019). Tools used for general bioinformatics analysis and protein-specific domains can be found at: http://www.ncbi.nlm.nih.gov/ (accessed on 14 April 2019) and http://www.ebi.ac.uk/Tools/ (accessed on 14 April 2019).

### 2.3. Generation of Transgenic Plants and Constructs

To generate constructs for production of transgenic plants in this study, the *ATL12* sequence was amplified using Phusion^®^ High-Fidelity DNA Polymerase from New England Bio-Labs (NEB, Ipswich, MA, USA). For Gateway entry cloning, Taq DNA polymerase was added to the raw PCR product to generate 3′ A-Overhangs. The resulting PCR products were purified, and entry clones were generated by recombination into the pCR™8/GW/TOPO vector (Thermo Fisher Scientific, Waltham, MA, USA), using the pCR^TM^8/GW/TOPO^®^ TA Cloning^®^ Kit (Thermo Fisher Scientific, Waltham, MA, USA). To generate overexpression constructs for *ATL12*, we utilized the destination vector pMDC 32(35S promoter--attR1-CmR-ccdB-attR2). After sequencing to confirm that the clone is in frame and oriented correctly, the recombinant construct was transformed into *Agrobacterium tumefaciens* strain (pGV3101) via chemical transformation using the floral dip method. The positive transformants with green expanded cotyledons and long hypocotyls were selected in 50 μg mL^−1^ hygromycin MS agar plates for 7 to 10 days. The T3 progeny from these transformants was used in this study. Transcript levels of *ATL12* were measured via quantitative RT-PCR in all over-expression lines to confirm results.

### 2.4. Subcellular Localization of ATL12 Protein

ATL12:GFP overexpression transgenic plants were constructed expressing an N-terminal ATL12-GFP fusion protein. For the ATL12:GFP transgenic plants, the *ATL12* sequence was amplified using Phusion^®^ High-Fidelity DNA Polymerase (New England Bio-labs). Entry clones were generated by recombination into the pCR™8/GW/TOPO vector, based on a previously published protocol [11]. The destination vector pMDC 43(35S-attR1-CmR-ccdB-attR2-GFP) was used to generate overexpression constructs. The construct sequence was confirmed, and then the recombinant construct was transformed into *Agrobacterium tumefaciens* strain (pGV3101). For subcellular localization in tobacco cells using confocal fluorescent microscopy, ATL12:GFP and the fluorescence plasma membrane marker pm-rk (plasma membrane marker fused with red ^M^cherry protein) were co-infiltrated into tobacco cells, the tobacco was then placed in the dark for two days for the fusion protein to be expressed and then the leaves were examined.

PCR-driven overlap extension was used to delete the transmembrane domain of *ATL12* to construct *ATL12ΔTM*, and three-step fusion PCR was conducted to create the 35S:*ATL12ΔTM*: GFP construct.

First step primers:

p*ATL12*-*ATL12*ΔTM:

5′-ATCCACCTTCATAAGCTGGTAATAGA-3′ (*pATL12-ATL12* forward)

5′-AGGTGAGAGTTCCGGTGATGATA-3′ (*ATL12ΔTM 1* reverse)

Second step primers:

5′-ATCACCGGAACTCTCACCTTTG-3′ (*ATL12ΔTM 2* forward)

5′-TGTTTTAGGATGGTGATTCGATGAG-3′ (*pATL12-ATL12* reverse)

Third step primers:

p*ATL12-ATL12ΔTM*:5′-ATCCACCTTCATAAGCTGGTAATAGA-3′(pATL12-ATL12 forward) and 5′-TGTTTTAGGATGGTGATTCGATGAG-3′ (*pATL12-ATL12* reverse)

The resulting PCR products were gel purified and cloned into a TOPO entry vector based on previous protocols [11]. Positive clones were recombined with pMDC32 (CD3-738) using LR Clonase Enzyme Mix and then transformed into *A. tumefaciens* strain GV3101.

35S:*ATL12ΔTM:GFP* agrobacterium GV3101 strain and fluorescence plasma membrane marker pm-rk (plasma membrane marker fused with red ^M^cherry protein) were co-transferred into tobacco cells. Free GFP was also infiltrated into tobacco leaves as a control. Fluorophores are visualized concurrently using a Nikon ECLIPSE Ti2 confocal fluorescent microscope (Nikon, Tokyo, Japan) equipped with an OptiGrid imaging system using FITC HYQ (Excitation: 460–500 nm; Emission: 510–560 nm) and TRITC HYQ (Excitation: 530–560 nm; Emssion: 590–650 nm) filters (Nikon, Tokyo, Japan). Images were generated and merged using NIS-Elements software Ver4.60.00 (Nikon, Tokyo, Japan).

### 2.5. Disease Assessment

To assess the susceptibility of Arabidopsis *ATL12* mutants to pathogen infection, the following experiments were performed. Arabidopsis seeds were placed in the cold room for three days, planted in soil, and then transferred to a growth chamber (22 °C day/19 °C night with 12 h of light per 24 h and 50% relative humidity). 21-day old seedlings were inoculated with powdery mildew and placed in the infection chamber with the same temperature and light conditions, except at 80% humidity. To qualitatively assess how the disease develops, we observed the appearance of inoculated leaves 7 days post-inoculation to check powdery mildew infection symptoms via trypan blue staining. Inoculated leaves were treated with ethanol before staining with trypan blue solution for 15 min. Leaves were harvested for staining at 6- or 7-days post-inoculation. Also, the number of spores per colony was determined to quantitatively assess the susceptibility of the mutants to pathogen infection. Approximately 0.5 g of seedlings were harvested at 7 dpi. 5 mL H_2_O was then added, and spores were released by vortexing for 30 s at maximum speed. For each sample, spore suspension solutions were filtered and added onto the eight 1 mm^2^ fields of a Neubauer-improved haemocytometer and spores were counted. Spore counts results were averaged and then normalized to the weight of seedlings. Each experiment was repeated 3 times. Statistical significance among samples was analyzed using one-way ANOVA followed by post hoc tests.

### 2.6. Histochemical Staining Assay

Transgenic Arabidopsis plants expressing p*ATL12*-GUS were generated and a histochemical staining assay was used to determine the p*ATL12*-GUS (β-Glucuronidase) expression pattern. *ATL12* sequence was amplified using Phusion^®^ High-Fidelity DNA Polymerase from New England Bio-Labs. Taq DNA polymerase was added to the raw PCR product to generate 3′ A-overhangs and the resulting PCR products were purified, and entry clones were generated by recombination into the pCR™8/GW/TOPO vector. To generate the overexpression constructs for *ATL12*, we utilized the destination vector pMDC 162 (-attR1-CmR-ccdB-attR2-GUS). Then the recombinant construct was transformed into *Agrobacterium tumefaciens* strain (pGV3101) using chemical transformation. The positive clones were selected in LB agar plates contained 100 μg mL^−1^ kanamycin and the frame and orientation were checked via sequencing. The floral dip method [23] was then used to generate overexpression transgenic plants and selection for the transgenics was performed using 1x MS agar plates contained 50 μg mL^−1^ hygromycin. The positive transformants with green expanded cotyledons and long hypocotyls were selected in hygromycin agar plates for 7 to 10 days. The T3 progeny from these transformants were used in this study. The screened p*ATL12*-GUS seeds were placed into 1× MS liquid culture and grown to specific developmental stages. The seedlings and tissues were then collected and stained overnight at 37 °C in GUS staining buffer. GUS expression was observed directly under a dissecting microscope.

### 2.7. Reactive Oxygen Species (ROS) Detection via DAB Staining Assay

To determine if *ATL12* expression was related to ROS production during pathogen infection, Arabidopsis Col-0, the T-DNA insertional mutants of *ATL12* and *ATL12* overexpression lines were grown under normal conditions (22 °C day/19 °C night with 12 h of light per 24 h and 50% relative humidity) for 3 weeks. Plants were then inoculated with powdery mildew and placed in a growth chamber with the same temperature and light conditions, except at 80% humidity. At 7 days post-inoculation, at least 5 leaves were removed from each plant and placed in a 24-well microtiter plate. 1 mL of the 10 mM Na_2_HPO_4_ DAB staining solution (50 mg DAB, 45 mL sterile H_2_O, 25 μL Tween 20 (0.05% *v*/*v*) and 2.5 mL 200 mM Na_2_HPO_4_, pH 3.0) was added to the leaf or leaves. The volume in each well was adjusted to ensure that the leaves were immersed in the DAB solution. The brown precipitate formed by the DAB reacting with the hydrogen peroxide was observed by light microscopy, and photographs were taken using an imaging system.

### 2.8. Quantitative RT-PCR and RT-PCR

To determine the temporal expression patterns of *ATL12*, we performed qRT-PCR with chitin-treated seedlings. Arabidopsis seedlings were growing in MS liquid medium in 50 mL Falcon tube for 14 days. We immersed the 14-day old wild type Arabidopsis seedlings with MS liquid medium contained 100 mg/mL chitin and then collected tissues at several time points. (1 h, 2 h, 4 h, 8 h, 16 h, 24 h). To determine if expression levels of *ATL12* were regulated by the classical plant defense pathways mediated by salicylic and jasmonic acids, we treated wild-type Arabidopsis seedlings with water, chitin, salicylic acid (SA) (2 mM), or jasmonic acid (JA) (100 μM) for two hours. The seedlings from the different treatments were then collected, frozen in liquid nitrogen and stored at −80 for qRT-PCR analysis. Total RNA was isolated from frozen tissues using TRizol Reagent (Invitrogen^®^, Carlsbad, CA, USA) according to the manufacturer’s protocol. RNA samples were treated with RQ1 DNase (Promega, Madison, WI, USA). Trace amounts of genomic DNA were removed by digestion with the Turbo DNA-free™ system (Ambion, Austin, TX, USA). First-strand cDNA synthesis was primed with an oligo (dT)15 anchor primer and cDNA was synthesized using a First-Strand Synthesis Kit (Amersham-Pharmacia, Rainham, UK) according to the manufacturer’s protocol. An aliquot of 1.5 µL of the first-strand synthesis reaction was used as a template for PCR amplification. To ensure that the sequence amplified was specific, a nested PCR was performed using 1 µL of a 1:50 dilution of the products synthesized in the first PCR reaction as a template. The RT-PCR, PCR, and nested PCR program consisted of: 3 min at 96 °C, 30 cycles of 30 s at 94 °C, 30 s at 65 °C, and 1 min at 72 °C, and a final extension step of 7 min at 72 °C. Amplified PCR fragments were visualized using 1% agarose gels.

The following gene-specific primers are used for RT-PCR:

*ATL12* (At2g20030): 5-CCCAACACACAGAGAGGTCGTC-9 (forward) and 5-GCTTCCACAATGACCTCCGA-9 (reverse);

*MAPK3* (At3g45640) 5′-TCTGCCTTTCCCTCTTCTCTC-3′ (forward primer), and 5′-GACAGCATTGACTCTGGCTCT-3′ (reverse primer);

AtRBOHD (At5G47910) 5′ATCAGTGCCGCATATTCTTTG-3′ (forward primer), and 5′-ATC

TTTCTTCCGAAGCACC TC-3′ (reverse primer);

AtRBOHF (At1G64060) 5′-AAACCAACACGCACCTTATTG-3′ (forward primer), and 5′-ATGAAATTGGCATTGCATTTC-3′ (reverse primer);

*ACTIN-2* (At3g18780): 5-AGCAGCTTCCATTCCCACAA -3 (forward) and 5- CATGCCA-TCCTCCGTCTTGA -3 (reverse).

Quantitative RT-PCR experiments were performed using a SYBR^®^ Green qPCR kit (Finnzymes, Espoo, Finland) with reactions at a final volume of 20 µL per well and using the cycle protocol recommended by the manufacturer. Samples were run in a DNA Engine Opticon^®^ 2 System instrument with PTC-200 DNA Engine Cycler and CFD-3220 Opticon™ 2 Detector (BioRad, Hercules, CA, USA). Gene-specific primers were designed using the Primer Express 2.0 program (Applied Biosystems, Foster City, CA, USA) and minimal self-hybridization and dimer formation of primers was determined using the Oligo 6.0 program (Molecular Biology Insights, West Cascade, CO, USA). Primers with annealing temperatures of 62 °C to 65 °C that amplified products with lengths of about 300 bp were selected and then verified for specificity by BLAST searches. The efficiency of amplification for each gene was calculated as recommended by the manufacturer (BioRad, Hercules, CA, USA).

The following gene-specific primers were used:

*ATL12* (At2g20030): 5-GAATTATGCCGTTACTGCGACC-9 (forward) and 5-ATTTTGG-CGTGTCGTGTTTAGG -9 (reverse);

*PR1* (At2g14610): 5-GCCTTCTCGCTAACCCACAT-3 (forward) and 5-CGGAGCTACGC-AGAACAACT-3 (reverse);

*MAPK3* (At3g45640): 5-TCATCATTCGGGTCGTGCAA-3 (forward) and 5-ACTTCCCAA-CTTCCCACGTC-3 (reverse);

*PDF1.2* (At5g44420):5-TGTTCTCTTTGCTGCTTTCGAC -3 (forward) and 5-TGCTGGGA-AGACATAGTTGCAT -3 (reverse);

*CERK1* (At3g21630) 5′-TCGAAACAGTTCTTGGCGGA-3′ (forward) and 5′-GGTTCTCGT-CCTGACCCATG-3′ (reverse);

*NPR1* (At1g64280) 5′-CCGGACCTGATGTATCTGCTC-3′ (forward) and 5′-GCGGTGTTGT-TGGAGTCTTTC-3′ (reverse);

JAZ1 (At1g19180): 5-GAGCAAAGGCACCGCTAATA-3 (forward) and 5-TGCGATAG-

TAGCGATGTTGC-3 (reverse);

*WRKY53* (At4g23810) 5′-TGGTGTCTTGTCGCTTCTCC-3′ (forward) and 5′-CAGAGATCA-GACGGGGATGC-3′ (reverse);

*Beta-ACTIN* (At3g18780): 5-AGCAGCTTCCATTCCCACAA -3 (forward) and 5- CATGCCA-TCCTCCGTCTTGA -3 (reverse).

Relative fold changes in transcript levels were determined using the double delta Ct Value (ΔΔCt) method. Data were acquired and analyzed using ANOVA followed by Turkey post hoc analysis. Three independent biological replicates were used in each experiment.

## 3. Results

### 3.1. Sequence Analysis of the Arabidopsis thaliana ATL12 Gene

In order to identify the *ATL12* gene, information was analyzed from two publicly available databases, the National Center for Biotechnology Information (NCBI) database, and The Arabidopsis Information Resource (TAIR) database. *ARABIDOPSIS TOXICOS EN LEVADURA 12* (*ATL12*), also named At2g20030 or *ATL2D*, belongs to the *ATL2* gene family, a group of really conserved C3HC4 RING-type protein with putative E3 ubiquitin ligase activity. The *ATL12* gene contains one exon and no introns, and the complete cDNA is 1505 bp long with its predicted ORF (1173 bp) encoding a 390 amino acid protein with a signal peptide, a transmembrane domain, and a RING-finger domain, as shown in Figure 1A. The complete cDNA and amino acid sequences of the *ATL12* are shown in Appendix A. The *Arabidopsis Toxicos en Levadura 2* (*ATL2*) gene family has 13 members and all appear to encode conserved RING-Zinc finger proteins with potential E3 ubiquitin ligase activity. They have been shown to be activated by chitin elicitors and play important roles in plant defense pathways [22]. All 13 members of the *ATL2* family contain RING domains and transmembrane domains, with only four members that have PEST domains. PEST domains are known to be associated with proteins that have short half-lives within the cell. Alignment of the ATL12 RING Zinc-finger domain amino acid sequence with the other *ATL2* gene family members showed that the consensus sequence for this group of RING proteins is: C-X2-CL-X-E-X7-R-X2-P-X-C-X-H-X-FH-X2-C-X-D-X-W-X6-CP-X-C, where X is any amino acid, as shown in Figure 1B.

### 3.2. Tissue Expression Pattern of ATL12

To investigate where and when *ATL12* is expressed at the tissue level, we generated transgenic Arabidopsis plants expressing the native promoter of *ATL12* driving a beta-glucaronidase (GUS) fusion in transgenic plants (p*ATL12*-GUS) and performed a histochemical staining assay to determine the *ATL12* expression pattern. Twenty of the screened T3 p*ATL12*-GUS seeds were placed into 1x MS liquid culture and grown to specific developmental stages (2 days, 3 days, 4 days, 5 days, 1 week, 2 weeks, 3 weeks, and 1 month). The results of this study are shown in Figure 2. The results showed that *ATL12* express at germinating seed and cotyledons during early embryogenesis (Figure 2a,b) and the root, leaf and stem of growing seedlings (Figure 2c–i). In addition, ATL12 is also found to express in flowers (Figure 2k,l).

### 3.3. ATL12 Localizes to the Plasma Membrane

Since ATL12 contains a transmembrane domain, we wanted to determine the subcellular localization of ATL12. To do this, a construct was made containing a C-terminal fusion of the *green fluorescent protein (GFP)* to the *ATL12* gene under the control of CaMV 35S promoter. *35S-ATL12-GFP* and the plasma membrane organelle fluorescence marker *PM-rk* were co-infiltrated into *Nicotiana benthamiana* (tobacco) leaf cells, and the tobacco leaves were placed in the dark for two days. The GFP and rk Fluorophores were then visualized in the leaf cells concurrently using confocal fluorescent microscopy. When visualized, the 35S-ATL12-GFP construct co-localized with the PM-rk marker PIP2 in the tobacco cells. This confirmed its localization to the plasma membrane as shown in Figure 3, top row. Additionally, a 35S-GFP construct was used as a negative control for GFP localization to the nucleus and cytosol (Figure 3, bottom row). Fluorescence was detectable within the nucleus in the tobacco cells expressing the 35S-GFP construct and the GFP signal could be seen in the nucleus (Figure 3, bottom row, white arrow). To determine if the transmembrane domain of ATL12 is responsible for its localization at the plasma membrane, an ATL12-ΔTM-GFP construct lacking the transmembrane domain was made and infiltrated into tobacco leaves. The ATL12-ΔTM-GFP protein was observed in the cytosol and in the nucleus (Figure 3, middle row, white arrow) but not at the plasma membrane, suggesting that the transmembrane domain of ATL12 is necessary for its localization at the plasma membrane.

### 3.4. ATL12 Expression Is Induced by Chitin Treatment

To determine the expression pattern of *ATL12* after the chitin treatment we used histochemical staining of transgenic plants containing the *ATL12* nature promoter fused with GUS. Four-week old transgenic leaves, expressing p*ATL12*-GUS were immersed in 1 mg/mL crab shell chitin (CSC) for chitin treatment. Distilled water was used as a control. As shown in Figure 4A, *ATL12* expression is highly induced after treatment with chitin for 2 h compared with the water control. To further determine the timing of *ATL12* expression after chitin treatment, two week old Col-0 wild type Arabidopsis were treated with 100 mg/mL chitin at different time points and the expression levels of *ATL12*, *the defense marker gene MITOGEN-ACTIVATED PROTEIN KINASE 3 (AtMAPK3)* and the reference gene beta-actin were determined using qRT-PCR. Wild-type Col-0 plants were treated with 100 mg/mL chitin and tissue samples were harvested at four early time points: 1 h, 2 h, 4 h, and 8 h post-treatment and at two late infection time points; 16 h and 24 h. The result of qRT-PCR is shown in Figure 4B. *MAPK3* expression is highly induced at the very early time point and is continuously induced all the time. *ATL12* is highly induced by chitin treatment and at the very early time point (2 h), then expression started to drop at 4 h and quickly drops off to normal levels at 16 h, which suggested that ATL12 is involved with chitin elicitor triggered immune response.

### 3.5. Mutants of atl12 Are More Susceptible to Golovinomyces Cichoracearum Infection

To determine if *ATL12* plays a role in Arabidopsis defense against powdery mildew infection, we obtained three T-DNA insertion mutant lines of *atl12* from the Arabidopsis Biological Resource Center (ABRC, Ohio State University) with stock numbers SALK_201056C SALK_066923C, and SALK_0950303C. The T-DNA insertion position of the three mutants and the confirmation of all three homozygous T-DNA insertion mutant lines were shown in Appendix A. Three T-DNA insertional mutants lines (*atl12*, *atl12a*, and *atl12b*), Col-0 wild type, and *NahG* (a hyper-susceptible mutant) were planted and 21-day old seedlings were inoculated with powdery mildew. To assess how the disease develops, we observed the appearance of inoculated leaves 7 days post-inoculation and determined the extent of powdery mildew infection symptoms via trypan blue staining as shown in Figure 5A,B. After powdery mildew infection, the *atl12* mutants’ appearance showed more powdery mildew development and spread compared to Col-0. While the overexpression line of *ATL12* showed little to no development and growth of the powdery mildew pathogen across the leaf surface (Figure 5A,B). In addition, the number of spores per colony was determined to quantitatively assess the susceptibility of the mutants to pathogen infection. Compared to Col-0 wild type, T-DNA insertional mutants of *atl12*, *atl12a*, and *atl12b* all displayed higher levels of spore generation. To further confirm *ATL12’s* role in fungal defense, we generated *ATL12* over-expression transgenic line, 20–30 seeds of the T3 progeny from transformants were used in the disease assessment assay. As shown in Figure 5C, we noticed that overexpression of *ATL12* has fewer spores generated in response to fungal infection compared to Col-0 WT, which suggests that over-expression of *ATL12* increases the resistance of Arabidopsis to fungal infection. Taken together, these results indicate that *ATL12* has a positive correlation with resistance to powdery mildew infection.

### 3.6. Chitin-Induced ATL12 Expression Is Linked to NADPH Oxidase AtRBOHD/F-Driven ROS Production

*ATL12* expression is highly induced after two hours of chitin treatment, suggesting that *ATL12* may be involved in the early events leading to chitin elicitor triggered immunity. Reactive oxygen species (ROS) are very important molecules produced in plant cells early in the infection process when plants are facing pathogen invasion or during abiotic stress that causes significant local cell damage [1,13,24]. Since the ROS response is crucial for disease resistance, and to determine if *ATL12* expression is related to ROS production during pathogen infection, Col-0, atl12 T-DNA insertional mutants and an *ATL12* over-expression line were inoculated with powdery mildew or treated with 100 mg/mL of chitin. Infected leaves of each mutant were double stained with diaminobenzidine (DAB) to visualize hydrogen peroxide (H_2_O_2_) and with Coomassie Blue to detect fungal hyphae and conidiophores. Compared to Col-0 wild type, all three *atl12* T-DNA insertional mutant lines showed less ROS production (Figure 6A(a)). To determine if ATL12 plays a role in the induction of ROS, leaves of Col-0, the insertional mutants of *atl12* and the *ATL12* over-expression line were detached and treated with 100 mg/mL of chitin for two hours. As shown in Figure 6A(b), ROS production showed no significant differences among Col-0 wild type, *atl12* T-DNA insertional mutant lines, and the overexpression lines without chitin treatment. While mutants of *atl12* (*atl12*, *atl12a*, and *atl12b*) displayed less ROS generation after chitin treatment compared to Col-0 wild type, the overexpression lines of *ATL12* had significantly more ROS generation, (Figure 6A(c)). These data suggest that *ATL12* may be responsible for ROS production at early timepoints for effective defense against powdery mildew.

Previous studies have shown that chitin receptor-mediated ROS production is linked to the *Arabidopsis thaliana* respiratory burst oxidase homolog D and F genes (*AtRBOHD/F*), two NADPH oxidases in Arabidopsis [13,25]. Our previous results have shown that *ATL12* may be responsible for ROS production in early defense responses against powdery mildew infection. To confirm if AtRBOHD expression is induced at the early timepoints. Wild-type Col-0 plants were treated with 100 mg/mL chitin and tissue samples were harvested at four early time points: 1 h, 2 h, 4 h, 8 h, 16 h and 24 h. *AtRBOHD gene expression* was determined using qRT-PCR. The result is shown in Figure 6B. *AtRBOHD* is highly induced by chitin treatment and at the very early time point (1 h), the expression reached a peak at 2 h and then started to drop at 4 h and 8 h. At 24 h after chitin treatment, *AtRBOHD* expression reached to another peak. Then to determine if *ATL12-*mediated ROS production is related to *AtRBOHD/F* expression, T-DNA insertion mutants, overexpression line, and Col-0 wild type plants were treated with 100 mg/mL chitin for 2 h or treated with MS medium as mock control, and expression of *ATL12*, *MAPK3*, *AtRBOHD*, *AtRBOHF*, and *ACT2* were monitored via RT-PCR. The *AtRBOHD, AtRBOHF*, and *MAPK3* expressions have no significant difference among *atl12* mutants, overexpression lines and wild type. As shown in Figure 6C, *AtRBOHD* and *AtRBOHF* expression is impaired in all three *atl12* mutant lines, while both *AtRBOHD/F* are upregulated in the *ATL12* overexpression line, suggesting that *ATL12* expression may be responsible for *AtRBOHD/F* expression.

To investigate the detailed relationship between *ATL12* and *AtRBOHD/F* induced ROS production, insertional mutants of *atrbohd* and *atrbohf* were tested to determine if loss of *AtRBOHD/F* affected *ATL12* expression*. atrbohd* and *atrbohf mutants* are verified by PCR based on methods mentioned previously, the result is shown in Figure 6D. Under normal condition, there are no significant differences of *ATL12* and *MAPK3* mRNA expression in *atrbohd/f* mutants, as shown in Figure 6E. *AtRBOHD/F* and *MAPK3* expression were significantly decreased in *atrbohd/f* mutants after chitin treatment, as shown in Figure 6E, while the expression of *ATL12* was not affected in either *atrbohd* or *atrbohf* mutants. This suggests that chitin induced *ATL12* expression may link to the AtRBOHD/F-related signal transduction pathway. Taken together, these results suggest that *ATL12* is involved in *AtRBOHD/F*-mediated defense responses and that the expression of *ATL12* and *AtRBOHD/F* is needed for effective defense against powdery mildew infection.

### 3.7. ATL12 May Act Downstream of Chitin-Mediated MAP Kinase 3 (MAPK3) Signaling

One of the major events occurring during the early chitin-induced immune response is the activation of a *MAPK* cascade [13]. Our results show that *ATL**12* expression is highly induced at early time points after chitin treatment, which suggests that *ATL12* may be involved in some of the earliest events during the chitin elicitor induced immune response. To determine if *ATL12* expression is linked to the activation of the *MAPK* cascade, several T-DNA insertional mutants in known chitin-mediated signaling pathway genes (*cerk1*, *mapk3*, *wrky53*) were obtained and tested. The confirmation of all homozygous T-DNA insertion mutant lines is shown in Appendix A. Each mutant line was treated with 100 mg/mL chitin for 2 h and expression levels were determined via RT-PCR and qRT-PCR. *ATL12* expression was downregulated in both *cerk1* and *mapk3* after treatment with chitin, while *ATL12* expression was not affected in *wrky53* (Figure 7A). The expression of *CERK1* was downregulated in *atl12* and *mapk3.* Additionally, *MAPK3* expression was impaired in both *atl12* and *cerk1* and slightly upregulated in *wrky53* (Figure 7A). As expected, *WRKY53* expression was downregulated in both the *cerk1* and *mapk3* mutants. Taken together these data indicate that *ATL12* is linked to the chitin-mediated signaling pathway through the action of the chitin induced MAPK3 cascade.

Previous research suggested that knock out mutants in *CERK1* completely lost the ability to induce ROS and activate the MAPK cascades in chitin-induced defense [12]. Our results suggest that *ATL12*-induced ROS production is necessary for *AtRBOHD/F* expression during this response. To confirm the relationship between *ATL12* expression, ROS production and MAPK cascade activation, *cerk1*, *mapk3*, and *atl12* were treated with 100 mg/mL chitin for 2 h and expression levels were monitored via qRT-PCR. *ATL12* expression was downregulated in *cerk1*, *mapk3*, *atrbohd*, and *atrbohf* (Figure 7B). The expression of *CERK1* was downregulated in both *atl12* and *mapk3* and *MAPK3* expression was impaired in *atl12* and *cerk1* (Figure 7B). *AtROBHD* and *AtRBOHF* gene expression were both downregulated in *atl12*, *mapk3* and *cerk1*, suggesting that *ATL12* may function as a mediator between chitin-elicitor triggered immunity and AtRBOHD/F-mediated defense responses.

### 3.8. Influence of SA-, JA- and Chitin-Mediated Pathways on ATL12 Expression

The salicylic acid (SA)-mediated pathway is involved in plant defense against biotrophic pathogens, while jasmonic acid (JA, MeJA) is predominantly linked with defense against necrotrophic pathogens [26]. To determine if the expression of *ATL12* was not only induced by chitin but also involved in the SA-, and JA- mediated signaling pathways, we treated Col-0 plants with 100 mg/mL chitin, 100 μM MeJA or 2 mM salicylic acid for 2 h and monitored the expression of *ATL12*, and marker genes in each signaling pathway (*PDF1.2* (JA), *PR1* (SA), *MAPK3* (chitin)) using qRT-PCR. As shown in Figure 8A, the expression of *ATL12* is highly induced by chitin treatment and is also induced by both MeJA and SA treatment. *MAPK3* was highly induced by chitin treatment and was slightly upregulated in the SA treatment but expression of *MAPK3* was not affected by MeJA treatment (Figure 8A). As expected, *PR1* was highly induced by SA treatment and slightly upregulated above control levels in the MeJA treatment and *PDF1.2* was highly induced by JA treatment and was strongly downregulated in the SA treatment (Figure 8A). Interestingly, the expression of *both PR1* and *PDF1.2* showed no significant difference between the chitin treatment and normal conditions. From these data, it indicated that *ATL12* expression is highly induced by chitin treatment and is also enhanced by JA and SA treatment, which suggests that *ATL12* may be involved in crosstalk between SA-, JA- and chitin-mediated signaling pathways.

Since *ATL12* is not only induced by chitin but is also induced by SA and JA treatment, we wanted to further investigate how *ATL12* contributes to SA and JA signaling during chitin-triggered immune responses. Col-0 wild type, *atl12* and overexpression *ATL12* seedlings were treated with 100 mg/mL chitin for 2 h and the expression levels of three marker genes, *MAPK3* (chitin), *Nonexpressor of PR genes 1* (*NPR1*, SA marker) and *Jasmonte zim-domain protein 1* (*JAZ1*, JA marker) were determined. As shown in Figure 8B,C, *atl12* showed less *NPR1* and *JAZ1* expression compared to Col-0 wild type, while overexpression of *ATL12* increased the expression of *NPR1* and *JAZ1*. These data further suggest that *ATL12* is involved in both SA- and JA-mediated signaling during chitin triggered immune responses.

## 4. Discussion

### 4.1. ATL12 Is a Putative E3 Ubiquitin Ligase That Is Involved in Defense Responses against Fungal Pathogens

The plant 26S proteasome/ubiquitination is crucial in response to stress and in degrading misfolded proteins [18]. The 26S proteasome system is also key in regulating signals in plant innate immunity [27,28,29]. Great attention has been given to the plant 26S proteasome/ubiquitination system in the past decade, especially to the discovery and characterization of E3 ubiquitin ligases [17,18]. In Arabidopsis, there are over 450 predicted proteins that contain one or more RING domains [30], which are predicted to have E3 ubiquitin ligase activity. *ATL12* belongs to *ATL2* gene family and it appears that many of the members of the *ATL2* gene family most likely encode RING-H2 zinc-finger domain proteins with at least six cysteines and three histidine amino acids. In addition, all proteins in the ATL2 family have at least one transmembrane domain and 4 of them have a PEST domain. While most of the functions of all the *ATL2* family genes are still unknown, there are some studies shown that *ATL* family genes are involved in chitin elicitor triggered immune response or defense responses against fungal pathogens [26]. For example, *ATL2* is one gene of a multigene *ATL* family, which encoded highly conserved RING-H2 zinc-finger proteins that may function as E3 ubiquitin ligases, and *ATL2* expression is highly induced with chitin [22]. Some early chitin-response genes and known pathogenesis-related genes such as *NPR1* are induced in constitutive expression of the *ATL2* gene mutants [23]. *ATL9*, one gene belongs to the *ATL2* gene family that encoded a RING domain contained zinc finger protein with E3 ubiquitin ligase activity has been proven to be involved in Chitin- and NADPH oxidase-mediated defense responses [11,31].

Here, we have shown that *ATL12* encodes a RING-H2 zinc-finger domain protein that may functions as RING-type E3 ubiquitin ligase and is highly induced by chitin treatment and fungal infection. Our results have shown that *ATL12* is involved with resistance against *G. cichoracearum*. *ATL12* mRNA expression is highly induced by chitin elicitor treatment at early time points (2 h). In addition, *ATL12* is induced by JA and SA treatment but displays differential expression patterns. Taken all together, it appears that *ATL12* may be involved in the early response of chitin elicitor triggered defense response and engaged in SA or JA mediated defense response pathways. However, the detailed molecular mechanisms of ATL12 in chitin elicitor triggered defense responses and SA or JA mediated defense response pathways remain unknown. Our future studies will focus on the *ATL12’s* possible interacting partners within chitin-mediated signaling pathway using the high-throughput sequencing or bimolecular fluorescence complementation-based cDNA library screening to help us better understand its role in plant innate immunity and defense responses against fungal pathogens.

### 4.2. ATL12 Is Involved in Both NADPH Oxidase-Mediated and Chitin Mediated Defense Responses

Reactive oxygen species (ROS) are very important molecules produced in plant cells when plants are facing pathogen infection or abiotic stress. High concentrations of ROS are harmful to plant species and cause significant oxidative damage or even cell death if not cleared by the ROS scavenging system [32,33,34]. However, ROS when present at low levels within cells are recognized as important signaling molecules, which involved in normal plant growth [35] and responses to biotic and abiotic stresses [34,36]. In *Arabidopsis thaliana*, ROS generation mediated by the activation of NADPH oxidase (RBOH) is well studied and has been shown to be involved with fungal defense [37,38] and abiotic stresses tolerance [39,40,41].

Our results showed that mutants of *atl12* have less ROS generation after chitin treatment compared to Col-0 wild type, while overexpression of *ATL12* has more ROS generation. At the same time, we observed that the expression of *AtRBOHD* and *AtRBOHF* is downregulated in the mutant of *atl12* and upregulated in overexpression lines. Our previous research showed that both *atrbohd*, and *atrbohf* mutants were more susceptible than wild type to *G. cichoracearum* and relatively low levels of hydrogen peroxide accumulation were detected [11,34,38]. Considering these results together, we believe that the expression of *ATL12* is related to *AtRBOHD* and *AtRBOHF* medicated ROS generation after chitin treatment. The *ATL12* expression involved NADPH oxidase-mediated signaling is required for defense against *G. cichoracearum* infection. Moreover, to further study the relationship between *ATL12*, with *AtRBOHD/F*-induced ROS production, we treated the mutants of *atrbohd* and *atrbohf* with 100mg/mL chitin and RT-PCR indicated that the expression of *ATL12* is not affected in either the *atrbohd* or *atrbohf* mutant, suggesting that *ATL12* may act upstream of *AtRBOHD/F*-related signal transduction. Previous studies have shown that in knock out (KO) mutants of *cerk1*, the KO mutant completely lost the ability to respond to chitin and activate chitin-associated MAPKs, generate ROS, and activate defense-related gene expression [13,14,15]. In our results, we noted that the expression of *ATL12* is downregulated in the KO mutant of *cerk1*. Taken together, these data suggest that *ATL12* is integral in both chitin-mediated defense responses and NADPH-mediated ROS production. *ATL12* may serve as a connection between these two defense-signaling pathways with ATL12 acting as one of the key regulators. The detailed mechanism linking *ATL12* expression with chitin signaling and NADPH-mediated ROS production has yet to be elucidated and will be the subject of future investigation.

### 4.3. Possible Role of ATL12 in Hormone, NADPH Oxidase-Mediated and Chitin- Mediated Defense

In this study, we noted that *ATL12* is highly induced by chitin treatment and fungal invasion and it also appears that *ATL12* may act downstream of chitin-mediated signaling pathway, via the MAP kinase 3 (*MAPK3*) signaling cascade. Additionally, we showed that *ATL12* is linked to ROS production and RT-PCR indicated that *ATL12*-related ROS production is dependent on expression of both *AtRBOHD* and *AtRBOHF*. Moreover, *ATL12* appears to act upstream of *AtRBOHD/F*-related signaling in Arabidopsis. qRT-PCR results showed that *ATL12* expression is affected by JA and SA treatment, suggesting that *ATL12* may act in SA- and JA-mediated defense signaling [42]. However, further studies are required to unravel the complete role of *ATL12* in hormone-mediated defense responses. Considering the data collectively, we hypothesize that *ATL12* may be involved in crosstalk or coordination between the JA- and SA-mediated, NADPH oxidase-mediated, and chitin-mediated signaling pathways. A possible model demonstrating the role ofATL12 in mediating information flow between these diverse pathways is shown in Figure 9. The figure was created with BioRender. Overall the data from the current study indicate that *ATL12* may be involved in a broader spectrum plant defense response. Future studies to determine ATL12’s interacting partners that link this E3 ligase with the JA- and SA-induced defense responses, NADPH oxidase-mediated signaling and the current model of chitin-mediated signaling will provide us with a more complete picture of the links between chitin signaling and a robust defense response against fungal pathogens.

## Figures and Tables

**Figure 1 jof-07-00883-f001:**
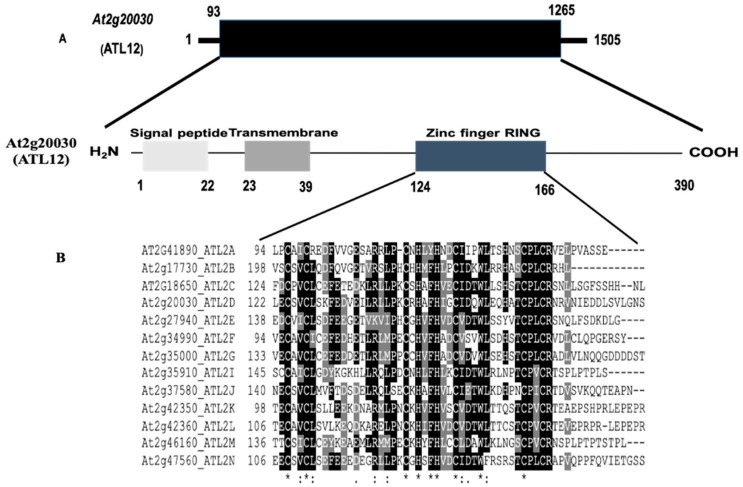
Sequence and domain analysis of *ATL12*. (**A**) The structure of genes and predicted domains for ATL12. One transmembrane domain (TM); and a C3HC4 RING Zinc-finger domain. (**B**). The consensus sequence for this group of RING zinc-fingers is C-X2-CL-X-E-X7-R-X2-P-X-C-X-H-X-FH-X2-C-X-D-X-W-X6-CP-X-C, where X is any amino acid. * indicates perfect alignment; : indicates a site exhibiting strong similarity; . indicates a site exhibiting weak similarity.

**Figure 2 jof-07-00883-f002:**
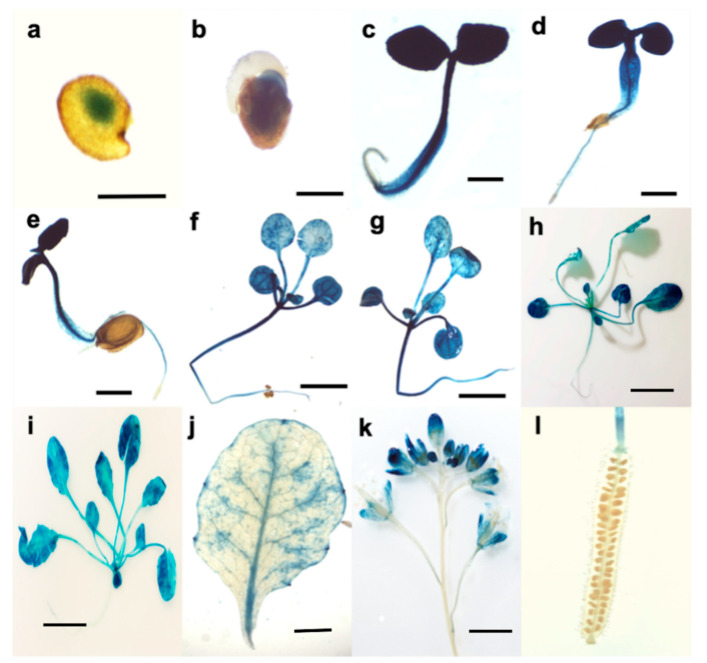
Tissue expression pattern of *ATL12*. Histochemical analysis of *ATL12* native promoter activity in different Arabidopsis tissues. p*ATL12*-GUS activity was detected in 1-day-old germinating seeds, (**a**) 2-day-old seedlings (**b**), 3-day-old seedlings (**c**), 4-day-old seedlings (**d**), 5-day-old seedlings (**e**), 6-day-old seedlings (**f**), one week-old seedlings (**g**), two-week-old seedlings (**h**), three-week-old seedlings (**i**), four-week-old leaf (**j**), Flowers (**k**), and siliques (**l**). Scale bar represents 1 mm.

**Figure 3 jof-07-00883-f003:**
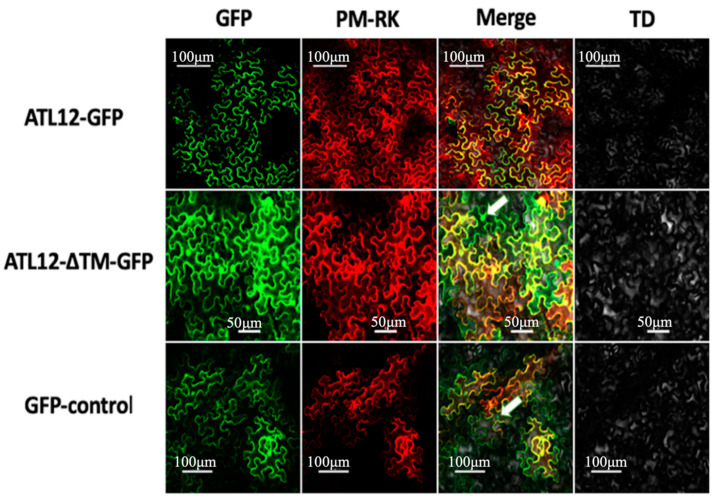
ATL12 protein is localized to plant plasma membrane. Sub-cellular localization of the ATL12-GFP protein. GFP and plasma membrane-M^cherry^ marker (PM-RK) were used as specificity controls for cytosol proteins PM-localized proteins. FITC HYQ channel (Excitation: 460–500 nm; Emission: 510–560 nm) was used to detect GFP signal and TRITC HYQ (Excitation: 530–560 nm; Emission: 590–650 nm) filters were used for PM-RK signal.

**Figure 4 jof-07-00883-f004:**
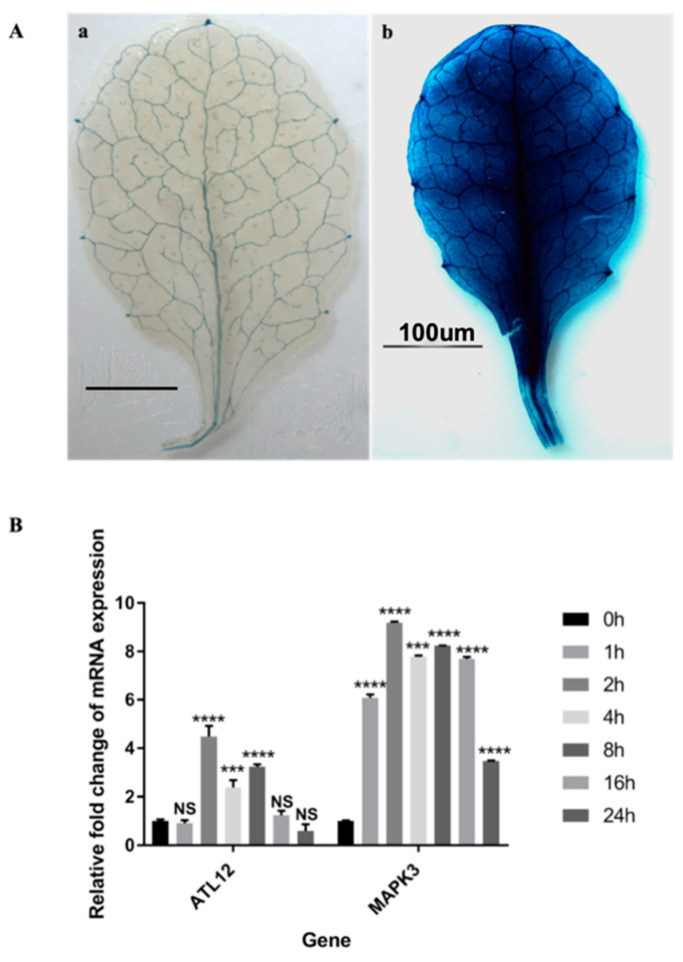
Expression pattern of *ATL12* after chitin treatment. (**A**): Histochemical staining assay of pATL12-GUS. 4 weeks old transgenic plant leaves express pATL12-GUS in leaves were treated with water (**a**) or with (**b**) 1 mg/mL CSC treatment for 2 h, and then subject to GUS staining. (**B**). qRT-PCR analysis of *ATL12* mRNA and *MAPK3* expression in response to chitin at different time points. Asterisks indicate statistically significant differences between the samples treated and untreated, according to the One-way ANOVA analysis and multiple comparison post-Tukey’s test. The experiments were repeated three times. **** indicates *p* < 0.0001, *** indicates *p* < 0.001, and NS indicates not significant. The black scale bar indicates 100 um.

**Figure 5 jof-07-00883-f005:**
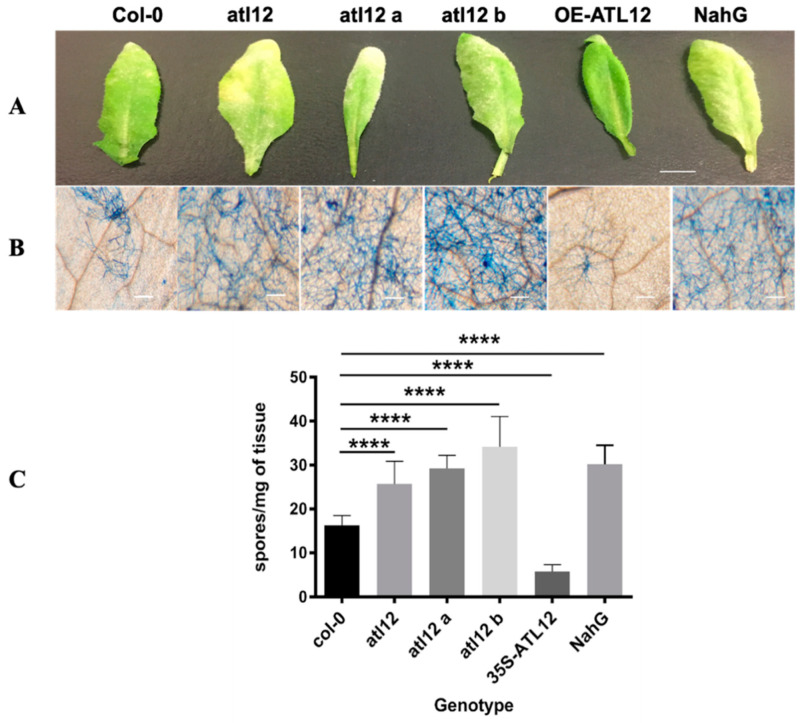
Phenotypic analysis of *Col-0 wild-type* (WT), *atl12* mutants and overexpression (OE) *35S:ATL12* plants in response to powdery mildew infection. (**A**). Leaf appearance of wildtype, *atl12* mutants, overexpression (*35S:ATL12*) seedlings and NahG transgenic plants (highly susceptible to fungal infection) 7d after powdery mildew infection. The white scale bar indicates 1 mm. (**B**). Trypan blue staining of the fungal structures on leaves 7days after fungal infection. (**C**). Quantitative phenotypic analysis of each seedling. One-way ANOVA followed by Tukey’s test was used to analyze the differences among treatments. Experiments were repeated three times. **** indicates *p* < 0.0001.

**Figure 6 jof-07-00883-f006:**
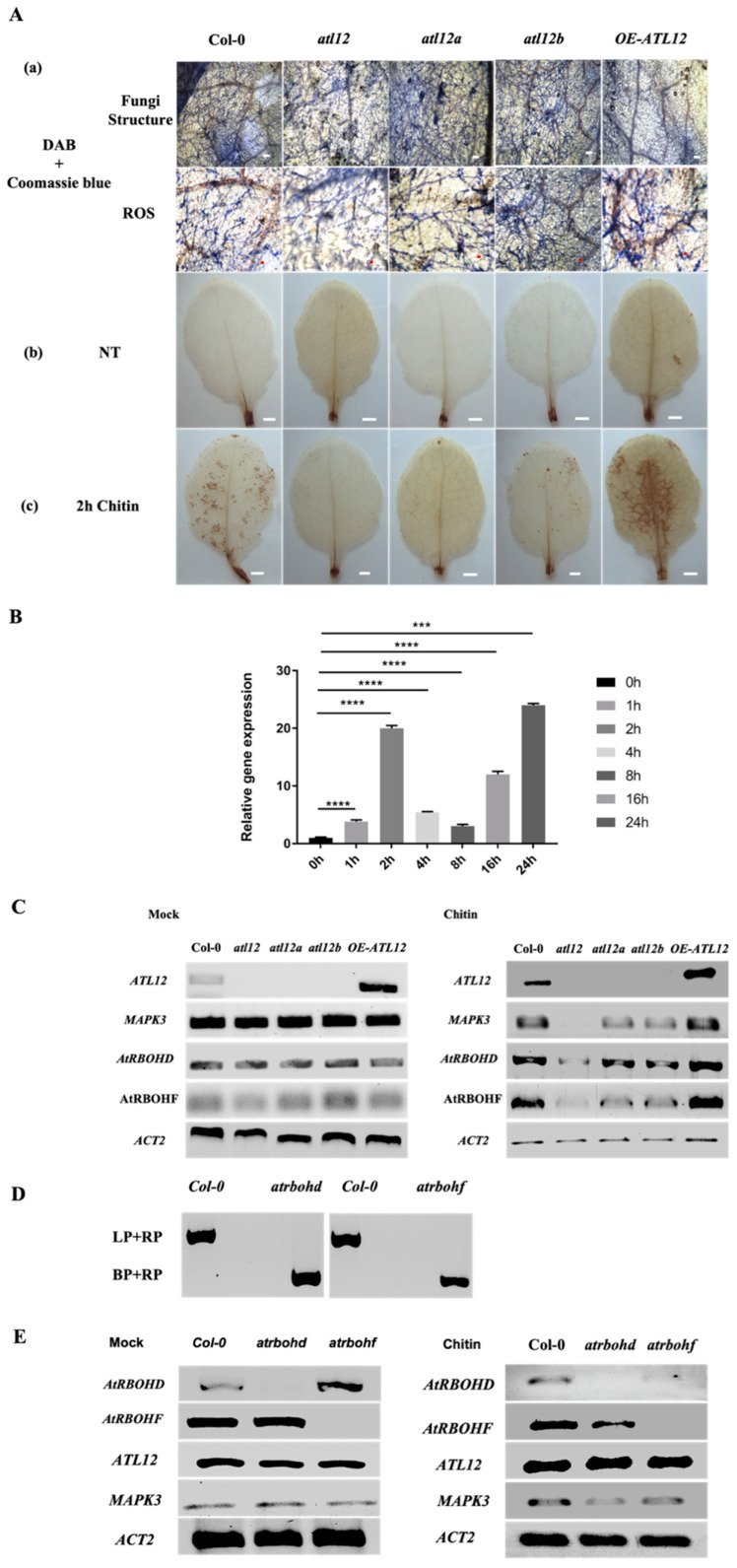
*ATL12* is responsible for *AtRBOHD/F*-dependent ROS production in response to chitin. (**A**). (**a**) DAB and Coomassie blue staining of plants in response to fungal treatment. Blue line indicated fungal hyphae, and brown precipitate indicated ROS. (**b**) DAB staining of different seedlings without chitin treatment; (**c**) DAB staining of plants treated with 100 mg/mL chitin for 2 h. Scale bar indicated 1 mm. (**B**). Temporal expression level of AtRBOHD mRNA expression after chitin treatment via qRT-PCR. (**C**). RT-PCR analysis of *ATL12*, *MAPK3*, *AtRBOHD*, and *AtRBOHF* expression in wild-type, atl12 mutants and overexpression (OE) lines under normal condition(mock) and after chitin treatment. (**D**). Amplification of *ATL12* in genomic DNA from the wild type, *atrbohd and atrbohf*. LP and RP are mutant-specific gene primers. BP is border primer in T-DNA insertional chunk in mutant. (**E**). RT-PCR analysis of *AtRBOHD*, *AtRBOHF, ATL12*, and *MAPK3* expression in mutant *atrbohd*, *atrbohf*, and Col-0 plants under normal conditions (mock control) and after chitin treatment. Experiments were repeated three times. The white and red scale bar indicates 1 mm. *** indicates *p* < 0.001 and **** indicates *p* < 0.0001.

**Figure 7 jof-07-00883-f007:**
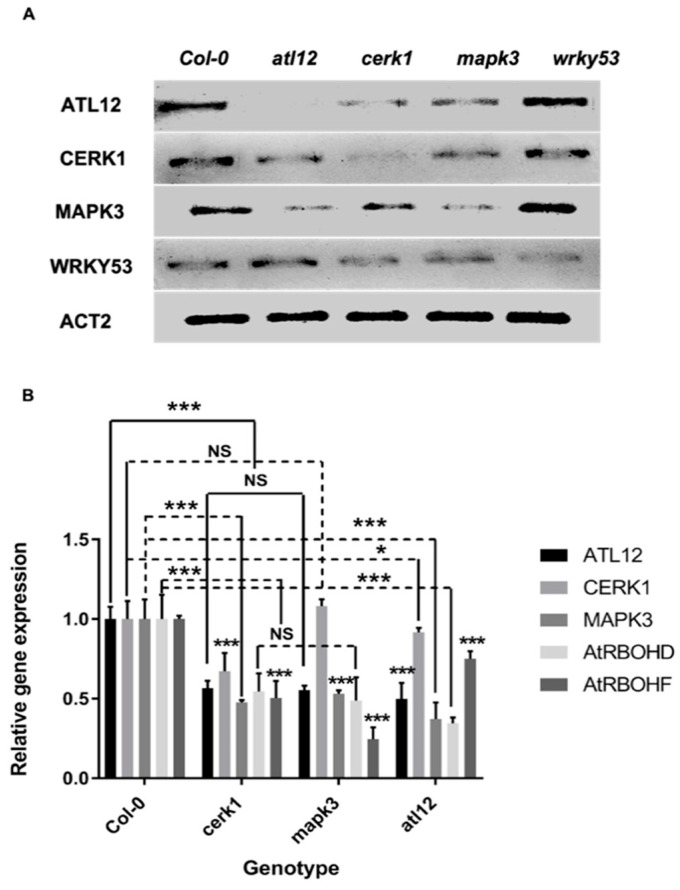
*ATL12* is involved with the chitin-mediated signaling pathway through *MAP kinase 3 (MAPK3)* cascades. (**A**). RT-PCR analysis of marker gene expression in wildtype and mutants. (**B**). qRT-PCR analysis of *ATL12*, *CERK1*, *MAPK3*, *AtRBOHD*, and *AtRBOHF* expression in wildtype and mutants. Experiments were repeated three times. *** indicates *p* < 0.001; * indicates *p* < 0.05 and NS indicates not significant.

**Figure 8 jof-07-00883-f008:**
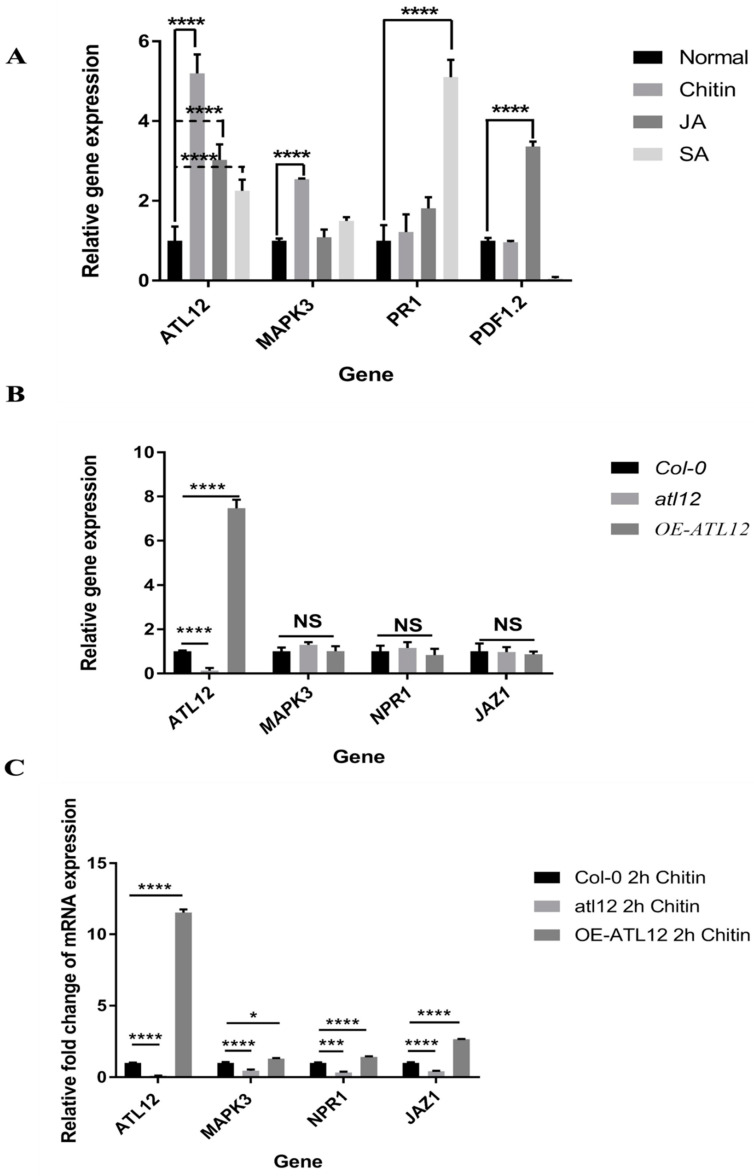
Influence of SA-, JA-, and Chitin-mediated pathways on *ATL12* expression (**A**). qRT-PCR analysis of marker gene expression in response to 100 mg/mL chitin, 100 μM MeJA, and 2 mM Salicylic acid for 2 h and ATL12, internal control beta-actin and marker genes for each signaling pathway were determined. (**B**). qRT-PCR analysis of marker gene expression under normal conditions. (**C**). qRT-PCR analysis of marker gene expression in response to 100 mg/mL chitin for 2 h and ATL12, internal control beta-actin and marker genes for each signaling pathway expression were determined. NS indicated no significance. Asterisks indicate statically significant differences between the samples treated and untreated, according to the One-way ANOVA analysis and multiple comparisons post-Tukey’s test. The whole experiments were repeated three times. **** indicated *p* < 0.0001, *** indicated *p* < 0.001, and * indicated *p* < 0.05.

**Figure 9 jof-07-00883-f009:**
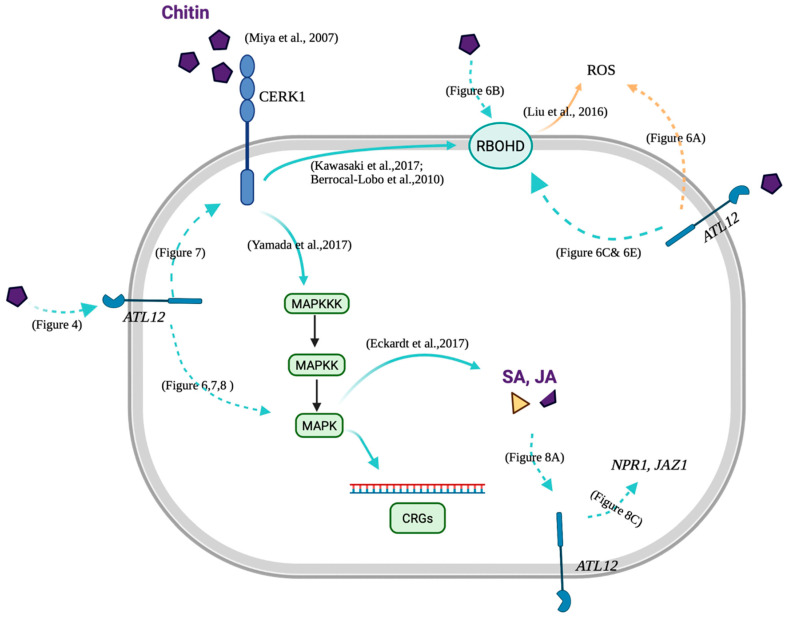
A possible model of ATL12 function in the hormone, NADPH oxidase-mediated and the chitin-mediated defense. ATL12 is at the downstream of chitin-mediated signaling pathway, through the MAP kinase 3 (MAPK3) cascade. Besides, ATL12 is related to ROS generation and may act at the upstream of ATRBOHD/F-related signaling transduction pathway. Moreover, ATL12 may be involved in the SA-, and JA-mediated signaling pathway [42]. The figure with created with BioRender.com with license agreement number DV2337X56K.

## Data Availability

Data is contained within the article or Appendix A.

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
