# Peer review of "Arabidopsis Toxicos en Levadura 12 (ATL12): A Gene Involved in Chitin-Induced, Hormone-Related and NADPH Oxidase-Mediated Defense Responses"

_jof, 2021, doi:10.3390/jof7100883_

Round 1
Reviewer 1 Report
In the current study, authors observed the continual expression of ATL12 gene in varied tissues using GUS reporter driven by ATL12 promoter and plasma membrane localization of ATL12. By inoculating atl12 and OE ATL12 plants with Golovinomyces cichoracearum, authors found ATL12 play positive roles in Arabidopsis to fight against fugal pathogen. DAB staining results showed that atl12 plants treated with chitin resulted in less ROS production when compared with wild-type plants. Using qRT-PCR, authors found Chitin, JA and SA induce ATL12 expression. In chitin-treated atl12 mutant, the induction of MAPK3, RBOHD and RBOHF by chitin were suppressed. However, some results cannot support the drawn conclusion. Thus major revision is highly recommended. My comments are as follows.
- In the chitin treatment experiment, what method was used to treat leaves? Dipping? Please add details in the method.
- Page 3. Yellow-highlighted region, reference should be listed.
- Page 5, fig. 1. B is missing in the bottom panel. In the middle panel, it is protein structure. So ATL12 shouldn’t be Italic.
- Page 5. Yellow-highlighted region, should be Italic.
- Page 8, fig. 4A. font size of a and b should be the same.
- Page 8. Yellow-highlighted region, ATL12 mRNA and MAPK3
- Page 9. Yellow-highlighted region, should be fig.5A and 5B.
- Page 10 fig. 5.A and 5B, two or more OE-ATL12 lines should be used.
- Page 12 fig.6B. RBOHD was induced at 1 hpi but ATL12 was highly expression at 2 hpi which is later than RBOHD shown in fig.4 B. How to explain that ATL12 acts upstream of RBOHD?
- Page 10, Yellow-highlighted region, I, II and III were used in fig. 6 but a and b were used in fig.4. Please make sure the same letters were used.
- Page 11, Yellow-highlighted region, …overexpression lines without chitin treatment.
- Page 11, green-highlighted region, ROS was induced by chitin but not ATL12.
- Page 12, fig.6A. It is hard to tell the fungus population trend. Spores counting results should be included.
- 6C the gel results in left and right panels should be comparable, but ACT2 level in right panel is much less than that in left panel. Please either give the results showing similar ACT2 expression level or include mock Col-0 in the right panel.
- Page 13, underlined region, font should be modified.
- Page 14, fig.7A, It is known that WRKY53 acts downstream of MAPK3. But what is the reason that, in atl12, MAPK3 has decreased level but WRKY53 not?
- Page 16, fig.8B, OE-ATL12 should originally have higher expression level of ATL12 even without chitin treatment. What is the fold change of ATL12 in OE-ATL12? Mock control should be included in this figure to indicate the real induced expression change of
- Page 17/21/22/23/ Yellow-highlighted region,please modify manuscript.

Author Response
Dear Reviewer,
Thank you for taking the time to review our manuscript and for your helpful comments and suggestions. They have improved the quality of the manuscript. We have addressed all your comments and concerns with the manuscript, and the exact corrections are listed below:
1. In the chitin treatment experiment, what method was used to treat leaves? Dipping? Please add details in the method.
Page 11 line 422, we modified the sentence [Four-week old transgenic leaves, expressing pATL12-GUS were immersed in 1 mg/mL crab shell chitin (CSC) for chitin treatment. Distilled water was used as a control].
Page 6 line 287:(methods and materials section) we updated the chitin treatment method and added the following sentence [Arabidopsis seedlings were growing in MS liquid medium in 50ml Falcon tube for 14 days. We immersed the 14-day old wild type Arabidopsis seedlings with MS liquid medium contained 100mg/ml chitin and then collected tissues at several time points].
2. Page 3. Yellow-highlighted region, reference should be listed.
Page 3 line 111: we added the reference [11,22].
Page 3 line 114: the yellow highlighted region came from our preliminary study (infection study of one atl12 mutant). We modified the [previous research] to [preliminary study].
3. Page 5, fig. 1. B is missing in the bottom panel. In the middle panel, it is protein structure. So ATL12 shouldn’t be Italic.
Page 8, Figure 1. We added the Letter B to the bottom panel. We modified the ATL12 format to italic in the middle panel.
Line 380: we made a change on Figure legend.
4. Page 5. Yellow-highlighted region, should be Italic.
Page 9 line 406: we modified the font format to italic.
5. Page 8, fig. 4A. font size of a and b should be the same.
Page 11 line 465: we modified the font size of a and b in Figure 4A.
6. Page 8. Yellow-highlighted region, ATL12 mRNA and MAPK3
Page 12 line 470: we added the following […..and MAPK] in Figure legend.
7. Page 9. Yellow-highlighted region, should be fig.5A and 5B.
Page 12 line 482: we updated the Figure S2.
line 490: this has been corrected in the manuscript. [Figure 4A and 4B are modified to Figure 5A and 5B].
8. Page 10 fig. 5.A and 5B, two or more OE-ATL12 lines should be used.
Page 13 Figure 5: We agree with the reviewer that further elaboration on this point using two or more OE-ATL12 lines would provide a stronger conclusion. However, there is no other overexpression line of ATL12 currently available in the ABRC stock center and we used the Agrobacterium based floral dip method (insertion of DNA at random sites) to generate our over-expression line of ATL12. The OE-ATL12 line indicated in Figure 5 is a combination of 20-30 seeds of T3 progeny from these transformants. We believe that getting two or more OE-ATL12 lines is feasible. However, given the cost and time involved to characterize each OE-ATL12 line (screening for the transformants and sequencing, etc.), we were not able to get more OE-ATL12 at this moment. But we have modified the following sentence in paragraph 1 of the results section 3.5 (Page 12 line 496): 20-30 seeds of the T3 progeny from transformants were used in the disease assessment assay.
9. Page 12 fig.6B. RBOHD was induced at 1 hpi but ATL12 was highly expression at 2 hpi which is later than RBOHD shown in fig.4 B. How to explain that ATL12 acts upstream of RBOHD?
Page 14 Figure 6B: We thank the reviewer for pointing this out to us. We agree that our conclusion that ATL12 may act upstream of RBOHD may be overstated and confusing to the readers. We have changed this conclusion [ATL12 may act upstream of RBOHD] to [Chitin-induced ATL12 expression is linked to NADPH oxidase AtRBOHD/F-driven ROS production] (Page 13 line 515 and Page 14 line 602). Our data show that AtRBOHD was induced at 1 hpi, 4hpi, and 8hpi, but also highly induced at 2h, 16h and 24h. Our thought is that AtRBOHD expression at early time points (1hpi) is mediated by other signaling pathways. There is still a missing link between CERK1-mediated recognition of chitin and ROS production as we mentioned in our discussion. Also, based on Figure 4C and 4E, AtRBOHD expression is decreased in atl12 mutants after chitin treatment, while ATL12 expression is not affected in the atrbohd mutant. Thus, we suggested that ATL12 is responsible for the induction of the ROS production. We acknowledge the regulation of signaling events is not a linear model. However, a more detailed relationship between ATL12 and AtRBOHD will require further evidence to demonstrate experimentally, which we also mentioned in the discussion section.
10. Page 10, Yellow-highlighted region, I, II and III were used in fig. 6 but a and b were used in fig.4. Please make sure the same letters were used.
Page 13 Figure 6A; Page 13 line 521, line 524 and line 528: We modified [I, II and III] to [a, b and c].
Page 14 Figure 6: We modified the Figure format.
11. Page 11, Yellow-highlighted region, …overexpression lines without chitin treatment.
Page 11 line 525: we added [without chitin treatment].
12. Page 11, green-highlighted region, ROS was induced by chitin but not ATL12.
Page 13 line 531, we modified [induce] to [be responsible for].
line 556: We added a sentence [under normal conditions, there are no significant differences between ATL12 and MAPK3 mRNA expression in atrbohd/f mutants, as shown in Figure 6E].
13. Page 12, fig.6A. It is hard to tell the fungus population trend. Spores counting results should be included.
Page 14 Figure 6A: We apologize if in our original Figure 6A one is unable to discern the fungal population trend. However, precise spore count results were included in Figure 5 using same treatment. In addition, the double staining method was used to show ROS production along with fungal infection levels. We observed that there was very little brown precipitate (ROS production) in the atl12 mutants. The DAB staining results, and fungal population trends were shown in Figure 6A(b) and (c) and Figure 5, respectively. For this reason, we chose not to make this suggested change.
14. 6C the gel results in left and right panels should be comparable, but ACT2 level in right panel is much less than that in left panel. Please either give the results showing similar ACT2expression level or include mock Col-0 in the right panel.
Page 14 Figure 6C: We understand that the reviewer expected to see either the results showing similar ACT2 expression levels or the inclusion of mock Col-0 in the right panel of the figure, in order to compare the gene expression between the mock and treatment group that might provide more information leading to a more robust conclusion. However, RT-PCR has its own limitations as a semiquantitative method, and it cannot accurately compare the gene expression between mock and treatment group. Although the ACT2 expression level appears different between the two conditions, we were able to see differential gene expression after chitin treatment and drew our conclusion by comparing the different genotypes under the same conditions. In addition, MAPK3, ATL12 and AtRBOHD mRNA expression in the Col-0 background under normal conditions (mock) and after chitin treatment were shown in Figure 4B and Figure 6B. Given time and cost constraints, we did not repeat the entire experiment or perform additional qRT-PCR assays. qRT-PCR would have been preferred if we compared gene expression between the mock group and chitin treatment group for all different genotypes.
15. Page 13, underlined region, font should be modified.
Page 14 line 576: we have modified the font in the underlined region, and figure legend.
Page 15 line 584: We have modified figure legend.
Page 15 line 596: We have modified the Figure S3.
16. Page 14, fig.7A, It is known that WRKY53 acts downstream of MAPK3. But what is the reason that, in atl12, MAPK3 has decreased level but WRKY53 not?
Page 15 Figure 7A: We thank the reviewer for pointing out that WRKY53 is known to act downstream of MAPK3. However, WRKY53 is not the only gene that acts downstream of MAPK3. Our data showed that in atl12 mutants, MAPK3 has decreased expression levels but WRKY53 did not, suggesting that the effect of ATL12 on MAPK3 expression might not be mediated by WRKY53. However, the detailed relationship does warrant further investigation as we mentioned in the discussion section. In our model, we only suggest that ATL12 may act downstream of chitin-mediated MAP kinase 3 (MAPK3) signaling.
17. Page 16, fig.8B, OE-ATL12 should originally have higher expression level of ATL12 even without chitin treatment. What is the fold change of ATL12 in OE-ATL12? Mock control should be included in this figure to indicate the real induced expression change of
Page 16 Figure 8B, We performed qRT-PCR on the mock control. There were no significant changes in MAPK3, NPR1, and JAZ1 mRNA expression under normal conditions. Also, OE-ATL12 did have higher expression levels of ATL12 under normal conditions. We apologize if our original Figure 8B did not show the mock control. We have modified the Figure 8 and included the mock control in Figure 8B.
18. Page 17/21/22/23/ Yellow-highlighted region, please modify manuscript.
Page 17 Figure 8B and 8C: We have updated the figure.
Page 18 line 661: We have updated the figure legend.
Page 4 line 11: We have updated the reference [11,22].
Page 6 line 295: We have corrected the error and the SA and JA concentrations were added (salicylic acid (SA; 2 mM), or jasmonic acid (JA; 100 μM )).
Page 22 line 890: We have modified reference format.
We thank you again for your time and suggestions. If there are any further questions, please feel free to contact me.
Reviewer 2 Report
The MS by Kong et al., identified that AtATL12 is a new component involved in chitin-triggered immune pathway in Arabidopsis. AtATL12 plays a positive role in mediating chitin-induced signaling and fungal defense. The expression of AtATL12 is upregulated by chitin treatment. Additional data also provide important information that AtTL12 might link t RBOHD/F-mediated ROS production. Overall, all the experiments in the MS are well done and the MS tis pretty well written which could be considered for publication. I only have a few minor comments.
- The current model regarding chitin perception in plants is the receptor complex by AtCERK1-AtLYK5/AtLYK4 in Arabidopsis and OsCERK1-CEBiP in rice. I would like to suggest the authors update the knowledge in the introduction.
- For the ease of reading, please revise the name of CERK1 from Arabidopsis and rice as AtCERK1 and OsCERK1, respectively.
- The data from Fig1A could be considered to put in the supplemental figures.
- The model figure is not suitable for publication and could be improved significantly.
Author Response
Dear Reviewer,
Thank you for taking the time to review our manuscript and for your helpful comments and suggestions. They have improved the quality of the manuscript. We have addressed all your comments and concerns with the manuscript, and the exact corrections are listed below:
1. The current model regarding chitin perception in plants is the receptor complex by AtCERK1-AtLYK5/AtLYK4 in Arabidopsis and OsCERK1-CEBiP in rice. I would like to suggest the authors update the knowledge in the introduction.
Page 2 line 66: we have updated the information about AtCERK1-AtLYK5/AtLYK4 in Arabidopsis and OsCERK1-CEBiP in rice in introduction section.
2. For the ease of reading, please revise the name of CERK1 from Arabidopsis and rice as AtCERK1 and OsCERK1, respectively.
Page 2 line 73 and line 75: we modified CERK1 to AtCERK1 or OsCERK1.
3. The data from Fig1A could be considered to put in the supplemental figures.
Page 8 line 378: we removed Figure 1A and added to Supplement Figure S1.
Page 8 line 367: we added the following sentence [The complete cDNA and amino acid sequences of the ATL12 are shown in Supplement Figure S1].
Page 19 line 747: we added the sentence [The figure was created with BioRender].
4. The model figure is not suitable for publication and could be improved significantly.
Page 20 line 755: we updated the Figure 9.
Page 22 line 890: we modified the reference format.
We thank you again for your time and suggestions. If there are any further questions, please feel free to contact me.
Round 2
Reviewer 1 Report
- Page 12, fig.6A. It is hard to tell the fungus population trend. Spores counting results should be included.
Page 14 Figure 6A: We apologize if in our original Figure 6A one is unable to discern the fungal population trend. However, precise spore count results were included in Figure 5 using same treatment. In addition, the double staining method was used to show ROS production along with fungal infection levels. We observed that there was very little brown precipitate (ROS production) in the atl12 mutants. The DAB staining results, and fungal population trends were shown in Figure 6A(b) and (c) and Figure 5, respectively. For this reason, we chose not to make this suggested change.
In Fig. 6, authors are aiming to let readers see the ROS deposition, but the problem is that In Fig.6A(a) and Fig.5B, inconsistence can be seen. In Fig.5B,it is so obvious that less fungus were detected on OE-ATL12 plants than that on Col-0, however, in fig. 6A(a) I can not see the same trend. It is stated by authors that OE-ATL12 plants are more resistant than Col-0, but data from Fig.6A(a) couldn’t support authors ’idea on this. Thus I highly recommend that whenever fungal growth were shown, fungal spores counting or other data indicating plants resistance should be included.
